# Quantum Monte Carlo simulations in the trimer basis: First-order transitions and thermal critical points in frustrated trilayer magnets

Lukas Weber,[1][*] Andreas Honecker,[2] Bruce Normand,[3,4]
Philippe Corboz,[5] Frédéric Mila[4] and Stefan Wessel[1]

**1** Institute for Theoretical Solid State Physics, JARA-FIT, and JARA-HPC,
RWTH Aachen University, 52056 Aachen, Germany
**2** Laboratoire de Physique Théorique et Modélisation, CNRS UMR 8089,
CY Cergy Paris Université, 95302 Cergy-Pontoise, France
**3** Paul Scherrer Institute, CH-5232 Villigen-PSI, Switzerland
**4** Institute of Physics, Ecole Polytechnique Fédérale de Lausanne (EPFL),
CH-1015 Lausanne, Switzerland
**5** Institute for Theoretical Physics and Delta Institute for Theoretical Physics,
University of Amsterdam, Science Park 904, 1098 XH Amsterdam, The Netherlands

[*] lweber@physik.rwth-aachen.de

## Abstract

The phase diagrams of highly frustrated quantum spin systems can exhibit first-order quantum phase transitions and thermal critical points even in the absence of any long-ranged magnetic order. However, all unbiased numerical techniques for investigating frustrated quantum magnets face significant challenges, and for generic quantum Monte Carlo methods the challenge is the sign problem. Here we report on a general quantum Monte Carlo approach with a loop-update scheme that operates in any basis, and we show that, with an appropriate choice of basis, it allows us to study a frustrated model of coupled spin-1/2 trimers: simulations of the trilayer Heisenberg antiferromagnet in the spin-trimer basis are sign-problem-free when the intertrimer couplings are fully frustrated. This model features a first-order quantum phase transition, from which a line of first-order transitions emerges at finite temperatures and terminates in a thermal critical point. The trimer unit cell hosts an internal degree of freedom that can be controlled to induce an extensive entropy jump at the quantum transition, which alters the shape of the first-order line. We explore the consequences for the thermal properties in the vicinity of the critical point, which include profound changes in the lines of maxima defined by the specific heat. Our findings reveal trimer quantum magnets as fundamental systems capturing in full the complex thermal physics of the strongly frustrated regime.

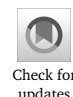

## 1 Introduction

Frustrated magnets typically possess a highly degenerate low-energy subspace, because no spin configuration can minimize every interaction term simultaneously. This degeneracy can be lifted by collective mechanisms leading to (quantum) order-by-disorder [1] or to a kalei-doscope of exotic disordered phases, including quantum spin liquids [2]. While some of this physics is captured by certain exactly solvable models [3–5], and the perturbative regimes close to them [6], many realistic models require numerical methods for their solution. For two-dimensional (2D) systems, these include exact diagonalization [7–9], the density matrix renormalization group [10–13], and recently developed tensor-network approaches [14–17]; although these are powerful methods, each has specific drawbacks. Another approach is the use of quantum Monte Carlo (QMC) methods, which using the stochastic series expansion (SSE) with directed loop updates are very efficient for quantum spin systems [18–20]. However, in the presence of geometric frustration, QMC typically suffers severely from the negative-sign problem [21–23].

Although the sign problem causes an exponential decrease in the efficiency of QMC, it is essentially basis-dependent, and for certain highly frustrated models it has been shown that a basis exists in which QMC simulations are exactly sign-problem-free (henceforth "sign-free"). Examples include fully frustrated two-leg $S = 1/2$ ladders in one dimension and the fully-frustrated bilayer (FFB) [Fig. 1(a)] in two [24–28]. In the FFB, the basic components are spin dimers, which are arranged on a square lattice and connected symmetrically by nearest-neighbor Heisenberg interactions, and the QMC sampling is performed in the local spin-dimer basis [24] (related QMC approaches were reported in Refs. [25, 29]). Similar sign-free bases exist not only for spin dimers but for arbitrary spin simplices, rendering generalizations of these models accessible by QMC [24, 25]. Further, each sign-free model is surrounded by an island in parameter space with a tolerable sign problem, which makes QMC useful also

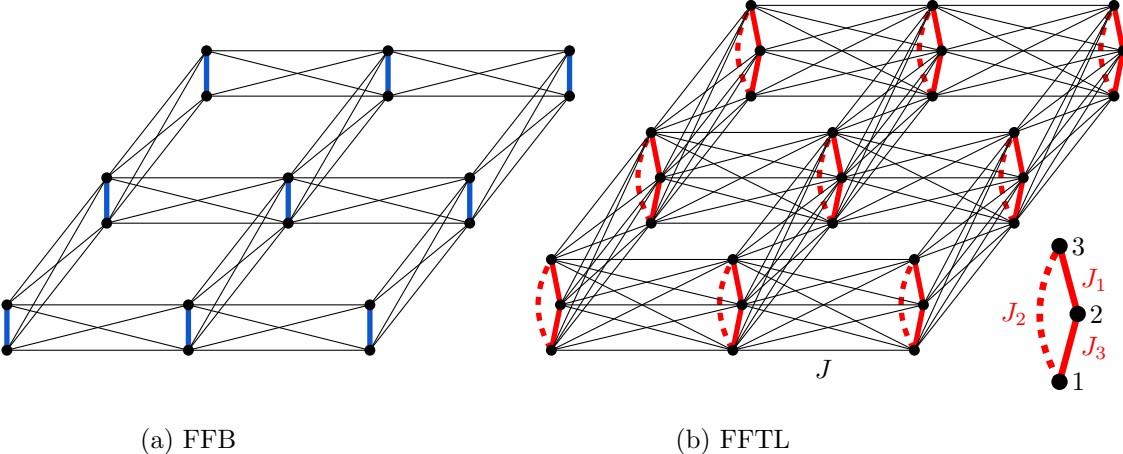

(a) FFB  (b) FFTL

Figure 1: The fully frustrated bilayer (a) and trilayer models (b). In the trilayer, the layers are labelled 1, 2, 3 from bottom to top. The spins of a trimer unit cell (inset) are coupled by intra-trimer interactions $J_1$, $J_2$, and $J_3$ (red) and to all spins in the nearest-neighbor unit cells by the same inter-trimer interaction, $J$ (black). The dashed red line denoting the coupling ($J_2$) between spins in the bottom (1) and top layers (3) of each trimer may differ from the other two couplings ($J_1 = J_3$).

for understanding related frustrated quantum magnets, such as the Shastry-Sutherland lattice (SSL) model [30] by its proximity to the FFB [31]. In the SSL model, if the dimer coupling is not too large, then using the dimer basis removes the sign problem completely in the zero-temperature limit. A similar effect occurs for frustrated quantum magnets in applied magnetic fields sufficiently far above the saturation field [32].

These recent advances in QMC methodology allow us to explore a number of important aspects of the thermodynamic properties in specific frustrated quantum magnets, and to trace their link to the ground-state properties. In the example of the FFB, the spin dimers shape the ground-state phase diagram, which is divided into a dimer-singlet quantum disordered and a dimer-triplet antiferromagnetic (AFM) phase, separated by a first-order quantum phase transition [33]. At finite temperature, it was observed that this transition persists, forming a first-order line, which terminates at a critical point [28], a scenario that has immediate parallels [34] to the liquid-gas [35], ferromagnetic [36], and Mott transitions [37].

In this work, we study a generalization of the FFB, the spin-1/2 Heisenberg trilayer with fully frustrated inter-trimer couplings (FFTL), illustrated in Fig. 1(b). The FFTL constitutes an extension of the fully frustrated three-leg ladder [38] to two dimensions, similar to the way in which the FFB is an extension of the fully frustrated two-leg ladder. In place of the dimers forming the basic components of the FFB, the FFTL is composed of trimer unit cells, whose internal frustration can be varied from zero at $J_2 = 0$ to maximal at $J_1 = J_2 = J_3$, and we will demonstrate that rich physics emerges from this internal degree of freedom. Finding this physics in a real material is likely to be a two-step process: although the geometry of the FFB has to date been realized only in a system with non-Heisenberg spin interactions [39], almost exactly the same physics is found in the compound $SrCu_2(BO_3)_2$ [34]. Similarly, while it is unlikely that a real material would possess the precise inter-trimer bonding of Fig. 1(b), several known quantum magnetic systems are based on triangular clusters with frustrated inter-triangle coupling [40–44], and thus are candidates to display some of the phenomenology obtained by varying the internal frustration of the FFTL.

To apply the SSE QMC method to the FFTL model, the sign-free basis is set by the local eigenbasis of the trimers. This basis consists of 8 states per unit cell, in place of the 4 states

in the spin-dimer unit cell, and simulating such large local bases usually comes at the cost of significant algorithmic complexity. To facilitate both this task and the implementation of specific physical operators, we therefore present a formulation of the SSE directed-loop update scheme that generalizes readily to arbitrary bases.

With these developments we investigate the thermal physics of the FFTL model based on sign-free QMC simulations. In particular, we are able to identify a first-order transition line emerging from a first-order quantum phase transition, which in the FFTL separates two distinct AFM ground states. Similar to the physics of the FFB and SSL [28, 34], we show that the first-order line of the FFTL is also terminated by a critical point, which we identify as belonging to the 2D Ising universality class. This phenomenology shares several similarities to the well known phase diagram of water, in which the line of first-order transitions in the pressure-temperature plane is terminated by a critical point belonging to the 3D Ising universality class. However, it was noted in Ref. [34] that in the 2D quantum magnets the specific heat exhibits two pronounced lines of maxima near the thermal critical point, in contrast to the single line of specific-heat maxima found in water. This behavior can be traced to the special case of the Ising model, whose $Z_2$ symmetry in fact enforces an exact symmetry between the two specific-heat lines in the magnetic-field-temperature phase diagram [34]. As we report below, varying the intra-trimer frustration in the unit cell allows us to (i) control the relative strengths of the two lines of maxima in the specific heat of the FFTL, and thus (ii) connect the generic "Ising-like" behavior observed to date in 2D quantum magnets to a more "water-like" type of behavior, featuring a single line of pronounced specific-heat maxima. Further, we connect the strongest water-like behavior, which occurs when the intra-trimer frustration is maximal, to an extensive entropy jump across the first-order quantum transition and hence to an enhanced "slanting" of the first-order transition line in the plane of "pressure" (coupling ratio) and temperature. Our observations thus establish the FFTL as a generic quantum spin model in which to explore the full complexity of the thermal physics uncovered in Refs. [28, 34], by the unbiased numerical technique of sign-free QMC simulations. Taken together, these provide us with the basis on which to understand the differences between the superficially distinct forms of thermodynamic behavior in certain quantum magnets and in water.

This article is structured as follows. In Sec. 2, we provide an overview of the FFTL model. In Sec. 3 we present the arbitrary-basis QMC method with which we simulate the FFTL in the trimer basis. This allows us in Sec. 4 to analyze the thermal physics of the FFTL, discussing in particular the first-order line, the thermal critical point, the behavior of the specific heat and of other quantities characterizing the critical fluctuations, and the effect of intra-trimer frustration on these properties. We conclude in Sec. 5 by discussing the implications of our results and their relation to other recent studies of thermal critical points in frustrated quantum magnets.

## 2 Model

In analogy to the well known FFB [25, 27, 28, 45–49] and the fully frustrated three-leg ladder [38], the FFTL is a square lattice of $S = 1/2$ spin trimers where all the sites in nearest-neighbor unit cells are connected by the same interaction, $J$, giving the Hamiltonian

$$H = \sum_\Delta H_\Delta + H_{\Delta,\Delta+\hat{x}} + H_{\Delta,\Delta+\hat{y}},$$

$$H_\Delta = J_1 \mathbf{S}_{\Delta,2} \cdot \mathbf{S}_{\Delta,3} + J_2 \mathbf{S}_{\Delta,3} \cdot \mathbf{S}_{\Delta,1} + J_3 \mathbf{S}_{\Delta,1} \cdot \mathbf{S}_{\Delta,2},$$

$$H_{\Delta,\Delta'} = J \sum_{i,j=1}^{3} \mathbf{S}_{\Delta,i} \cdot \mathbf{S}_{\Delta',j} = J \mathbf{S}_\Delta \cdot \mathbf{S}_{\Delta'}. \tag{1}$$

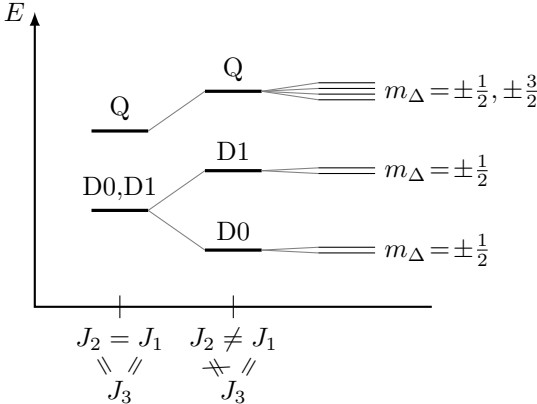

Figure 2: Spectrum of a single trimer with spin rotational symmetry. The 8 states belong to one quartet, Q, and two doublets, D0 and D1. The doublets are degenerate if and only if $J_1 = J_2 = J_3$.

The sum over $\Delta$ enumerates the trimer unit cells, $\mathbf{S}_{\Delta,i}$ is the $i$th spin of trimer $\Delta$, which belongs to the $i$th layer in Fig. 1(b), and $\mathbf{S}_\Delta = \mathbf{S}_{\Delta,1} + \mathbf{S}_{\Delta,2} + \mathbf{S}_{\Delta,3}$ is the total-spin operator of trimer $\Delta$. The unit cells are arranged as a square superlattice of $L \times L$ trimers.

The spectrum of a single trimer consists of 8 states that belong, due to the spin rotational symmetry of $H_\Delta$, to one quartet (denoted Q) or two doublets (D0 and D1), with respective eigenenergies $\varepsilon_Q$, $\varepsilon_{D0}$, or $\varepsilon_{D1}$ (Fig. 2). If $J_1 = J_2 = J_3$, the two doublets are degenerate, resulting for AFM interactions in a fourfold ground-state degeneracy. This is a direct consequence of the symmetry group $G = C_{3v} \times SU(2)$ at the degeneracy point: the irreducible representations of $G$ are tensor products of the irreducible representations of the two factors, and one obtains in this case a decomposition of the trimer Hilbert space into $A_1 \otimes \mathbf{4} \oplus E \otimes \mathbf{2}$. Away from this point, the doublet levels split and the ground state of a single trimer is only doubly degenerate. For $J_1 = J_3$, the eigenstates can be classified completely by the expectation values of the commuting operators

$$\mathbf{S}_\Delta^2 = l_\Delta(l_\Delta + 1), \tag{2}$$

$$(\mathbf{S}_{\Delta,1} + \mathbf{S}_{\Delta,3})^2 = l_{\Delta,13}(l_{\Delta,13} + 1), \tag{3}$$

$$S_\Delta^z = m_\Delta, \tag{4}$$

where $l_\Delta$ is the total-spin quantum number on trimer $\Delta$ and $l_{\Delta,13}$ the total-spin quantum number of spins 1 and 3 on this trimer. For convenience we restrict ourselves henceforth to this symmetric case (Fig. 1).

In the full Hamiltonian, the fully frustrated inter-trimer interaction, $J$, couples the total spins, $\mathbf{S}_\Delta$, of adjacent trimers. Because

$$[\mathbf{S}_\Delta^2, S_\Delta^\alpha] = 0, \tag{5}$$

$$[(\mathbf{S}_{\Delta,1} + \mathbf{S}_{\Delta,3})^2, S_\Delta^\alpha] = 0, \tag{6}$$

$$[S_\Delta^z, S_\Delta^\alpha] = i\epsilon^{z\alpha\beta}S_\Delta^\beta \neq 0, \tag{7}$$

$l_\Delta$ and $l_{\Delta,13}$ remain good (local) quantum numbers of the full model. However, $m_\Delta$ is no longer conserved.

A notable consequence of Eqs. (6) and (7) is that the matrix elements of the fully frustrated inter-trimer interactions in the "trimer" basis labelled by $l_\Delta$, $l_{\Delta,13}$, and $m_\Delta$ take the form

$$\langle l_\Delta, l_{\Delta,13}, m_\Delta | \mathbf{S}_\Delta | l'_\Delta, l'_{\Delta,13}, m'_\Delta \rangle = \delta_{l_\Delta, l'_\Delta} \delta_{l_{\Delta,13}, l'_{\Delta,13}} \langle m_\Delta | \mathbf{S}_\Delta^{S=l_\Delta} | m'_\Delta \rangle, \tag{8}$$

where $\mathbf{S}_\triangle^{S=l_\triangle}$ is a spin-$l_\triangle$ spin operator. The matrix elements therefore take the same values in the $S_\triangle^z$ basis as in an effective square-lattice AFM of mixed $S = 1/2$ and $S = 3/2$. The spin-$S$ Heisenberg AFM on a bipartite lattice can, however, be rendered sign-free [50], and consequently the inter-trimer Hamiltonian, $H_{\triangle,\triangle'}$, of the FFTL in the trimer basis, $|l_\triangle, l_{\triangle,13}, m_\triangle\rangle$, is also sign-free. The remaining intra-trimer part, $H_\triangle$, is diagonal in this basis and is thus sign-free as well. The trimer basis is therefore well suited to the construction of a general and efficient SSE QMC algorithm, which we present next.

# 3 Arbitrary-basis QMC

The SSE QMC method offers a highly efficient means of simulating quantum spin systems. It is based on expressing the quantum Boltzmann operator as an infinite-order high-temperature expansion with easily evaluated, positive coefficients [18]. This representation, which can be interpreted as a classical probability distribution, is then supplemented by the global directed-loop [19] class of updates that allow an efficient sampling of the probability distribution using the Markov-chain Monte Carlo approach.

In its original and most common formulation, the SSE is carried out explicitly in the single-spin $S^z$ basis. However, it is possible to use a different computational basis, generally at cost of additional algorithmic complexity, and alternative bases have been introduced predominantly for one of three reasons. First, models with a sign problem in the single-spin basis may be "cured" of it in a different basis. This is the case in dimerized Heisenberg models with fully frustrated interactions, as introduced in Sec. 1, where the enlarged dimer basis gives a fully sign-free problem, and the same situation arises in our current trimer-based model. Second, enlarged bases can also assist calculations for sign-free models that nevertheless pose a challenge to the efficiency of SSE QMC updates because of high energy barriers, as encountered in the case of Ising-type frustration [51–55]. Third, changes of basis can make observables that are very difficult to compute in one basis readily accessible in another; we will meet an example of this in Sec. 4, where the correlation functions of $\mathbf{S}_\triangle^2$ are in essence a four-spin observable.

While formulating the SSE in an alternative basis is straightforward, finding a directed-loop Monte Carlo update scheme that works efficiently in a given basis often is not. To date such update schemes have typically been implemented by hand-crafting loop processes describing the physics of the model to the chosen basis; this situation makes the exploration of new bases, for any of the three reasons listed above, a somewhat tedious task.

In this section we present a general scheme to automate this process. We start in Subsec. 3.1 by performing the straightforward step of arbitrary-basis operator decomposition, which we specialize to the FFTL for illustration. In Subsec. 3.2 we then detail an "abstract" loop-update scheme that can be applied to an arbitrary basis. The performance of this scheme is equivalent to a selected, physically motivated class of directed-loop updates, but the abstract-update approach generalizes readily to arbitrary bases. Particularly for the FFTL, where the trimer basis is rather large and the hand-crafted construction of a "proper" loop-update scheme (that encodes the full physics) is no longer transparent, the abstract-update approach is easy to apply and we will find in Sec. 4 that its efficiency is indeed sufficient to perform QMC simulations on length scales large enough to address the critical properties of the FFTL.

## 3.1 Operator decomposition

The starting point for the SSE is the decomposition of the Hamiltonian into non-branching operators (those which map each basis vector to a single basis vector, and not to a superposition

of different basis vectors). We express the FFTL model Hamiltonian in the form

$$H = \sum_{\langle \Delta, \Delta' \rangle} \tilde{H}_{\Delta,\Delta'}, \quad \tilde{H}_{\Delta,\Delta'} = H_{\Delta,\Delta'} + \frac{1}{4}(H_\Delta + H_{\Delta'}),$$

where the intra-trimer terms have been absorbed into the inter-trimer part, and decompose $H$ into the operators

$$H = \sum_{\substack{\langle \Delta, \Delta' \rangle \\ x_\Delta, x_{\Delta'} \\ y_\Delta, y_{\Delta'}}} \langle x_\Delta, x_{\Delta'} | \tilde{H}_{\Delta,\Delta'} | y_\Delta, y_{\Delta'} \rangle \, |x_\Delta, x_{\Delta'}\rangle \langle y_\Delta, y_{\Delta'}| =: \sum_\alpha h_\alpha, \tag{9}$$

denoting the sets of trimer-basis states $|l_\Delta, l_{\Delta,13}, m_\Delta\rangle$ for brevity by $|x_\Delta\rangle$ and $|y_\Delta\rangle$. The operators $h_\alpha$ of this decomposition are labelled by the bond $\langle \Delta, \Delta' \rangle$ and for notational simplicity we combine the summation indices into a single multi-index, $\alpha$. Using this decomposition, the partition function can be expanded into a form that is simple to evaluate because it contains only sums over non-branching operator strings (details may be found in Ref. [50]),

$$Z = \operatorname{Tr} e^{-\beta H} = \sum_{n=0}^\infty \sum_{\psi_0} \frac{\beta^n}{n!} \langle \psi_0 | (-H)^n | \psi_0 \rangle \tag{10}$$

$$= \sum_{n=0}^\infty \sum_{\psi_0} \sum_{\{\alpha_p\}_{p=1}^n} \frac{\beta^n}{n!} \langle \psi_n | \psi_0 \rangle \prod_{p=1}^n \langle \psi_{p-1} | (-h_{\alpha_p}) | \psi_p \rangle, \tag{11}$$

where $|\psi\rangle = |x_1, x_2, x_3, \cdots\rangle$ is the trimer-product basis of the full Hilbert space and the sum is over all $n$-sequences of $\alpha$. The sequence of intermediate states $\{\psi_1, \ldots, \psi_n\}$ is generated by the successive application of the non-branching operators $h_\alpha$ to the trace state $|\psi_0\rangle$.

To increase numerical efficiency, the sequences of $h_\alpha$ are in practice padded by identity operators so that they always have a fixed length, $\mathcal{L}$. This length is chosen adaptively to be sufficiently large that it has no measurable effect on the result. However, the padding procedure introduces a combinatorial factor that must be accounted for in the padded expansion (so that $n$ now denotes the number of non-identity operators contained in the operator string $\{\alpha_p\}$), leading to

$$\tilde{Z} = \sum_{\psi_0, \{\alpha_p\}_{p=1}^{\mathcal{L}}} \frac{\beta^n (\mathcal{L}-n)!}{\mathcal{L}!} \langle \psi_\mathcal{L} | \psi_0 \rangle \prod_{p=1}^{\mathcal{L}} \langle \psi_{p-1} | (-h_{\alpha_p}) | \psi_p \rangle =: \sum_{\mathcal{C}} W(\mathcal{C}). \tag{12}$$

Because the weights $W(\mathcal{C})$ are non-negative for the FFTL in the trimer basis, the quantum partition function can be interpreted as a classical probability distribution and different observables can be calculated using Markov-chain Monte Carlo methods.

## 3.2 Abstract directed-loop update

To sample from a Markov chain, we need an ergodic update for the configuration $\mathcal{C} = \{\psi_0, \{\alpha_p\}\}$ that fulfills the detailed-balance criterion. A combination of two types of update is used widely in the context of the SSE. First, the diagonal update inserts or removes diagonal operators (in the computational basis) locally in the string $\{\alpha_p\}$ [18]. Second, for the off-diagonal operators and the state $\psi_0$, global loop updates respecting the trace boundary condition on $\langle \psi_\mathcal{L} | \psi_0 \rangle \neq 0$ are employed [19, 56]. In general, directed-loop updates function as follows: starting from a given SSE configuration, two discontinuities are introduced at lattice site $i$ with operator index $p$. Referring to these discontinuities as the loop head and tail,

the head is propagated stochastically through the SSE operator string until it meets with the tail. At this point the two discontinuities can annihilate each other again, leaving a loop along which the spin configuration has changed.

The diagonal updates generalize readily to arbitrary bases, and so can be applied directly. For the directed-loop updates, however, the situation is more complicated. As noted above, approaches to date have tended to involve crafting the loops for each specific model by associating the possible discontinuities with physically intuitive operators. As an example, the operators $S^+$ and $S^-$ are used for spin systems in the conventional single-site $S^z$ basis. For local bases containing more than a single site, making this identification becomes increasingly difficult, because the number of basis states grows faster than the number of intuitive operators. An example from the dimer basis is the inclusion of the spin-difference operators, $\mathbf{D} = \mathbf{S}_1 - \mathbf{S}_2$, in addition to normal spin flips [24,26]. For the trimer basis, spin-difference operators alone are not sufficient, a result we illustrate by expressing in terms of spin components the operator that flips both $l_{\Delta,12}$ and $m_\Delta$ simultaneously,

$$\sum_{l_{\Delta,12},m_\Delta} |1/2,l_{\Delta,12},m_\Delta\rangle\langle 1/2,1-l_{\Delta,12},-m_\Delta| = \frac{4}{\sqrt{3}}\left[S_2^x(S_1^yS_3^y+S_1^zS_3^z-1)-S_1^x(S_2^yS_3^y+S_2^zS_3^z-1)\right]. \quad (13)$$

While it may be possible to simplify such an expression, and to introduce abbreviations for its operators, it is apparent that the language of physical loop operators becomes increasingly complex in larger cluster bases.

The alternative route we introduce is an abstract directed-loop update scheme that can be constructed canonically for any basis or model. As a general starting point, we associate the discontinuities at the loop head and tail not with operators but with bijective functions or "actions,"

$$a : X \to Y; \quad X,Y \subset \mathcal{B}, \quad (14)$$

which map the state before the discontinuity, $x$, to the state after the discontinuity, $a(x)$, both states being elements of the local (trimer or other arbitrary) computational basis, $\mathcal{B}$. In this picture, the equivalent of $S^+$ and $S^-$ for the single-spin $S^z$ basis would be the two actions $a^\pm : |m\rangle \mapsto |m\pm 1\rangle$, where $m$ denotes the $S^z$ eigenvalue. Note that the domain $X$ and image $Y$ of an action $a$ may not include all of the basis states, as is the case here for $a^+$ and the state $|m=S\rangle$.

The loop head moves through the configuration carrying one of these actions, and whenever it hits an operator $h_\alpha$ there are different ways to proceed, as represented in Fig. 3. The head can change its real-space position and direction by hopping to a different operator leg, $l$, or it can change the type of its discontinuity, meaning its action $a$. Depending on these choices, the operator is changed, $h_\alpha \to h_{\alpha'}$, after the head has passed. A way to bring this change of configuration into agreement with detailed balance is to demand that each junction, or scattering event, within the propagation process itself fulfills detailed balance.

Adopting the notation of actions $a$ and legs $l$, each operator scattering event can be described as a transition

$$\alpha, a_{\text{in}}, l_{\text{in}} \to \alpha', a_{\text{out}}, l_{\text{out}}. \quad (15)$$

If the operator before scattering was $h_\alpha = -w_\alpha |x_1,x_2\rangle\langle x_3,x_4|$, the new operator, $h_{\alpha'}$, is the one where first $x_{l_{\text{in}}}$ is replaced by $x'_{l_{\text{in}}} = a_{\text{in}}(x_{l_{\text{in}}})$, and in a second step the state at $l_{\text{out}}$ is set to $x''_{l_{\text{out}}} = a_{\text{out}}(x'_{l_{\text{out}}})$. The detailed-balance condition for this process is then

$$w_\alpha\, p(\alpha, a_{\text{in}}, l_{\text{in}} \to \alpha', a_{\text{out}}, l_{\text{out}}) = w_{\alpha'}\, p(\alpha', a_{\text{out}}^{-1}, l_{\text{out}} \to \alpha, a_{\text{in}}^{-1}, l_{\text{in}}). \quad (16)$$

Taken together, this condition and the normalization conditions for the transition probabilities produce a system of equations from which not all probabilities are determined uniquely. In

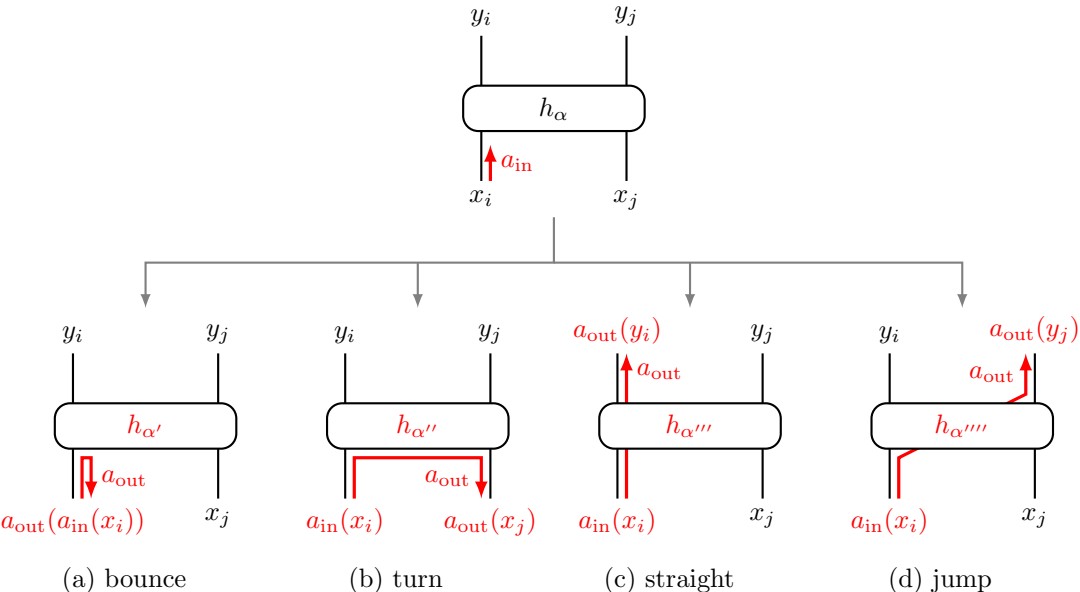

Figure 3: Abstract directed-loop updates. Whenever the loop head arrives at leg $l_{\text{in}}$ of an operator $h_\alpha$ carrying the action $a_{\text{in}}$, it may continue on a different leg, $l_{\text{out}}$, with a different action, $a_{\text{out}}$. Along the path of the head, the four states $x_i$, $x_j$, $y_i$, and $y_j$ defining $h_\alpha$ are changed by the application of $a_{\text{in}}$ and $a_{\text{out}}$.

practice, one typically chooses an optimal solution according to some heuristic criteria, such as the minimization of "bounce" processes (those for which $l_{\text{out}} = l_{\text{in}}$, $a_{\text{out}} = a_{\text{in}}^{-1}$, Fig. 3(a)) [19]. Finally, it is necessary to specify how the loop ends. If the head, carrying the action $a_h$, meets the tail, carrying the action $a_t$, and $a_h = a_t^{-1}$, the loop terminates. Otherwise, $a_t$ is replaced by the composition $a_h \circ a_t$.

So far, everything we have done is equivalent to the hand-crafted approach, with a slight change of nomenclature. The question for a given model is always how to choose the set of actions $\{a_k\}$ that the loop head can carry. In a sense, the most complete set of actions, $\{a_k, k = 1, ..., d-1\}$, that can be applied to a local basis, $\mathcal{B} = \{x_1, ..., x_d\}$ fulfills the property

$$\{a_k(x), \ k = 1, ..., d-1\} = \mathcal{B}\backslash\{x\} \qquad \forall x \in \mathcal{B}, \tag{17}$$

i.e., for all basis states $x \neq y$ there exists an action $a_k$ so that $a_k(x) = y$. In other words, the discontinuity at a loop head can change every state to every other state, given the right loop action. The identity transitions $a_k(x) = x$ can be excluded because they are in general not useful in changing the configuration.

Adding more actions to a complete set does not change the behavior of the loops, because all state discontinuities are already possible. Using a minimally complete set means that all hand-crafted implementations with fewer operators are included automatically as a solution in Eq. (16). Such an "incomplete" set of actions would not allow all discontinuities but does fulfill detailed balance; a solution equivalent to the incomplete set can be obtained within a complete set of actions by setting the probabilities leading to the "forbidden" discontinuities to zero. Thus for the abstract directed-loop update we use a complete set of actions.

There are clearly many choices for a set of functions fulfilling the completeness property from Eq. (17). However, because all possible state discontinuities are already allowed, the exact implementation of the set does not matter. Changing from one implementation to another merely constitutes a relabelling of the transition indices $\alpha, a_{\text{in}}, l_{\text{in}} \rightarrow \alpha', a_{\text{out}}, l_{\text{out}}$. As a result,

one may choose a set of functions on the basis of computational convenience, such as

$$a_k^{\mathrm{mod}} : x_n \mapsto x_{(n+k) \mod d}, \qquad k = 1, \dots, d-1, \tag{18}$$

which represents the cyclic permutations of the basis states (labelled here starting from zero, $n = 0, \dots, d-1$). When $d$ is a power of two, as in the present case, an even simpler set of functions exists in the form of the bitwise exclusive-or function (xor),

$$a_k^{\mathrm{xor}} : x_n \mapsto x_{\mathrm{xor}(n,k)}, \qquad k = 1, \dots, d-1. \tag{19}$$

One may ask how allowing all discontinuities, as opposed to allowing only a selected subset, impacts computational performance. In the example of an $S = 1$ Heisenberg model, the complete set of actions allows the head to interchange the states $|+1\rangle$ and $|-1\rangle$, which cannot be expressed using just the actions $a^+$ and $a^-$. It is well known that including $(S^\pm)^2$ loops, which would correspond to such discontinuities, does not increase the efficiency because of the $S_i^+ S_j^-$ structure of the bond Hamiltonian. Starting from a complete set of actions, one would therefore need to go beyond the bounce-minimization heuristic for optimal performance and search for solutions of Eq. (16) that discourage magnetization changes $\Delta m \neq \pm 1$ explicitly. Using such a solution, the behavior of the abstract directed-loop update is again equivalent to the standard $S^\pm$ one, other than some performance cost associated with larger data structures.

   The advantage of the abstract directed-loop update is in models and bases where knowledge about the optimal set of allowed discontinuities does not yet exist, such as the FFTL in the trimer basis. The abstract directed-loop update instead provides a canonical starting point, namely a complete set of actions with the bounce-minimization heuristic. This starting point is equivalent to any set of hand-crafted operators that allow all discontinuities and, in principle, can be optimized systematically.

   In our implementation, we use the $a^{\mathrm{xor}}$ actions and linear programming to minimize the probability of bounces in the solution of Eq. (16). While it is not always possible to avoid bounces completely, this heuristic alone was sufficient to achieve acceptable performance for the FFTL. More specifically, we were able to perform efficient simulations for systems of $L \times L$ trimers up to $L = 48$, at temperatures down to $T = 0.1 J_1$. For scans of the full $T$-$J$ plane, we restricted our simulations to systems with $L = 12$, this being sufficient to access the physics away from the critical point; close to the critical point, we performed finite-size scaling to capture the effect of the diverging correlation length.

## 4   Thermodynamics of the FFTL

We now present the results obtained by applying the abstract directed-loop QMC approach to examine the thermal properties of the FFTL. Phase diagrams in the plane of temperature and the ratio of inter- to intra-trimer coupling are presented in Figs. 4(a-d) for $J_1 = J_3$ with 4 different values of $J_2/J_1$. The phase diagram is obtained from the expectation value of the total spin, $\mathbf{S}_\triangle^2 = l_\triangle(l_\triangle + 1)$, on each trimer and the color scale indicates the mean of its associated quantum number, $\langle l_\triangle \rangle$.

   At low temperatures, $\langle l_\triangle \rangle$ approaches the value $1/2$ if the AFM intra-trimer couplings dominate, but sufficiently strong $J$ overrides this preference, optimizing the inter-trimer interaction terms by creating a ferromagnetic configuration on each trimer, causing $\langle l_\triangle \rangle$ to approach $3/2$. These effective $S = 1/2$ and $S = 3/2$ AFM ground states are lower in energy than any states of mixed $l_\triangle$, because the fully frustrated inter-trimer interactions favor neighboring trimers with the same $l_\triangle$. Thus they are connected directly by a level-crossing, or first-order quantum phase transition, occurring at one $J_2$-independent value of the inter-to-intra-trimer coupling ratio, $J_c/J_1$.

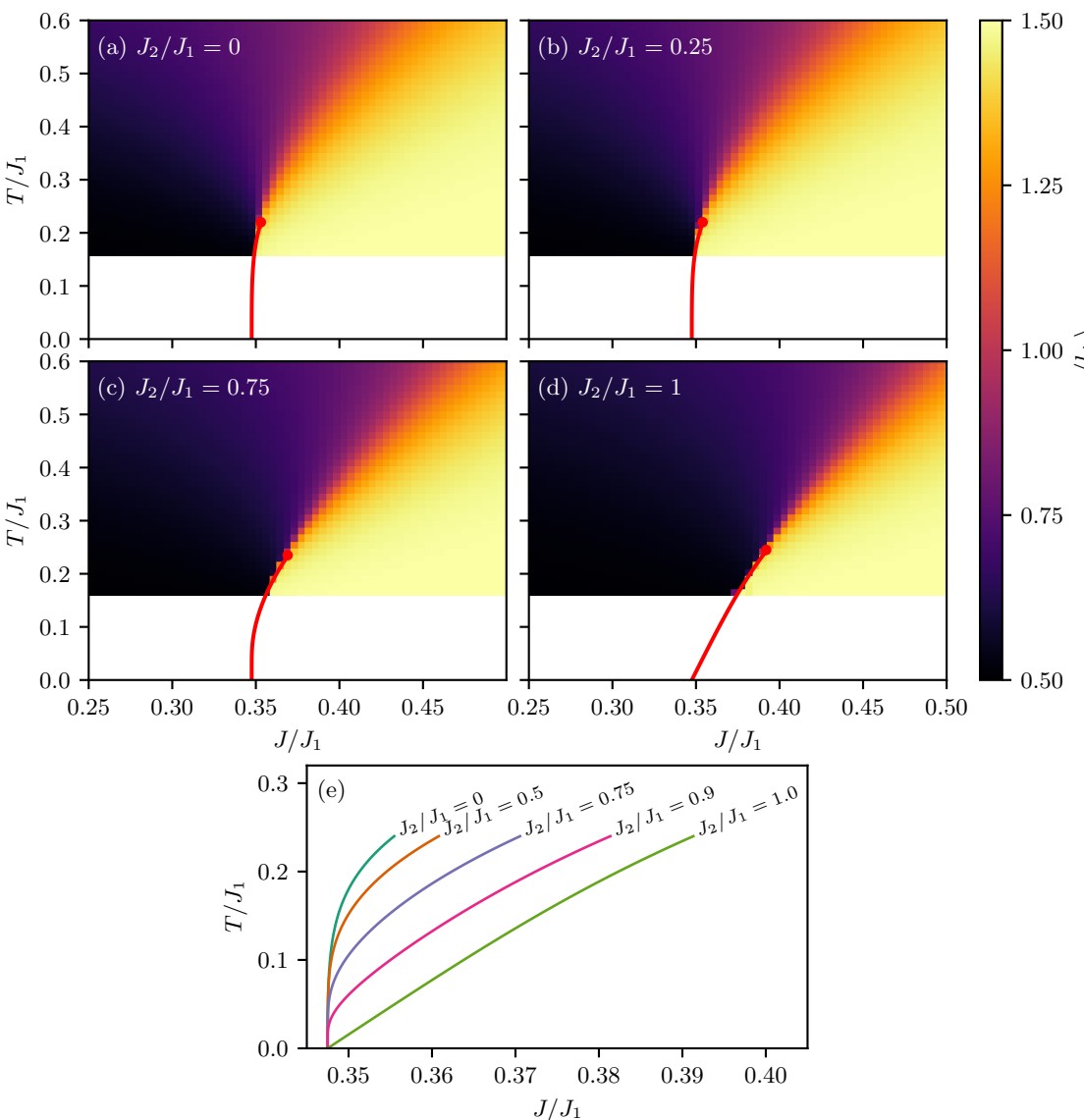

Figure 4: (a)-(d) Triangle total-spin quantum number, $\langle l_\Delta \rangle$, computed for the $L = 12$ FFTL model at different values of the intra-trimer coupling ratio $J_2/J_1$, with $J_1 = J_3$. Red lines show the lines of first-order transitions at finite temperature, calculated from the approximation developed in Subsec. 4.1, and red circles mark the critical point determined numerically in Subsec. 4.2. (e) Comparison of the shapes of the first-order lines obtained for different $J_2/J_1$ values from the approximation of Subsec. 4.1.

At low finite temperatures, this first-order transition persists, forming a line in the phase diagram. However, sufficiently high temperatures cause the transition to turn into a crossover, and hence each line in Figs. 4(a-d) terminates at a critical point. This behavior can be associated with the effects of thermal fluctuations into mixed-$l_\Delta$ states and we defer a detailed determination of the critical points to Sec. 4.2.

At a superficial level these results resemble the phase diagram of the FFB, but there are two major distinctions. First, the first-order transition takes place between two phases that both have a broken symmetry at $T = 0$. This is in contrast to the quantum disordered dimer-singlet ground state on one side of the transition in the FFB. Second, in the FFTL there is another independent control parameter in addition to the inter-trimer coupling, and it is clear from Figs. 4(a-d) that this $J_2/J_1$ parameter influences both the shape of the first-order line and the location of the critical point.

To investigate the physics of the FFTL phase diagram, we will first introduce some low-temperature approximations appropriate for gaining an analytical understanding of the first-order transition line (Subsec. 4.1). Because this approach does not allow us to extract the position of the critical point, in Subsec. 4.2 we perform detailed finite-size scaling to identify the critical point numerically and to discuss the influence of $J_2$ on the critical temperature. In Subsec. 4.3, we examine the specific heat and discuss its evolution with the shape of the first-order line. In Subsec. 4.4, we supplement this analysis by presenting results for the correlation length around the critical point, which allow us to compare its line of maxima with that of the specific heat.

## 4.1 First-order transition

With the goal of finding a straightforward means of explaining the shape of the first-order transition line in the low-temperature region, we use the locally conserved quantities of the FFTL model to obtain low-temperature approximations to the free energy on both sides of the transition, an approach used for the FFB in Ref. [28]. We start by rewriting the partition function using the local quantum numbers, $\{n_\Delta\} := \{(l_\Delta, l_{\Delta,13})\}$, to obtain

$$
\begin{aligned}
Z &= \mathrm{Tr} \exp\Big[-\beta\Big(\sum_\Delta H_\Delta + \underbrace{\sum_\Delta H_{\Delta,\Delta+\hat{x}} + H_{\Delta,\Delta+\hat{y}}}_{=:H_J}\Big)\Big] \\
&= \sum_{\{n_\Delta\}} e^{-\beta \sum_\Delta \varepsilon(n_\Delta)} \mathrm{Tr}\, P_{\{n_\Delta\}} e^{-\beta H_J} = \sum_{\{n_\Delta\}} \mathrm{Tr}\, e^{-\beta H(\{n_\Delta\})} =: \sum_{\{n_\Delta\}} e^{-\beta F(\{n_\Delta\})},
\end{aligned}
\tag{20}
$$

where $P_{\{n_\Delta\}}$ is a projection onto the subspace of the set of local trimer quantum numbers, $\{n_\Delta\}$, and

$$
H(\{n_\Delta\}) = \sum_\Delta \varepsilon(n_\Delta) + P_{\{n_\Delta\}} H_J P_{\{n_\Delta\}}.
\tag{21}
$$

Thus, $F(\{n_\Delta\})$ is the free energy of a system with a frozen configuration of trimer quantum numbers belonging to the partition function

$$
Z(\{n_\Delta\}) = \mathrm{Tr}\, e^{-\beta H(\{n_\Delta\})}.
\tag{22}
$$

In this way, the FFTL model can be regarded as a mixture of all possible ways to combine $S = 1/2$ and $S = 3/2$ trimer spins into a square-lattice AFM. Because $F(\{n_\Delta\})$ is an extensive quantity, the distribution in Eq. (20) will generally become sharp in the thermodynamic limit and at low temperature, and the system will choose the configuration $\{n_\Delta\}$ that minimizes $F(\{n_\Delta\})$.

Our first approximation is that $F(\{n_\Delta\})$ is minimized by a uniform configuration, $n_\Delta = n$. At low temperatures, this is expected because a system containing multiple domains of different $n_\Delta$ can decrease its energy by growing the single domain with the lowest energy per site. At high temperatures, however, "domain walls" become favorable due to their entropy, and lead to the appearance of the critical point. By neglecting domain walls, our approximation loses the ability to predict the end of the first-order line. Further, it is conceivable that domain walls between $l_\Delta = 1/2$ and $l_\Delta = 3/2$ could become energetically favorable even at $T = 0$, due to quantum fluctuation effects, although we will exclude this possibility *a posteriori* for the bare FFTL by comparing our approximate results to the QMC data.

For simplicity, we first consider $T = 0$, where $F(\{n_\Delta\})$ reduces to the ground-state energy, $E_0(\{n_\Delta\})$, of $H(\{n_\Delta\})$. As explained above, we assume

$$\min_{\{n_\Delta\}} E_0(\{n_\Delta\}) = E_0(n), \tag{23}$$

and restrict our considerations to three possible ground-state configurations with uniform $n_\Delta$, having total energies per trimer ($N_t$ denotes the number of trimers)

$$E_Q/N_t = \varepsilon_Q + J\varepsilon_{\text{AFM}}^{S=3/2}, \tag{24}$$

$$E_{D1}/N_t = \varepsilon_{D1} + J\varepsilon_{\text{AFM}}^{S=1/2}, \tag{25}$$

$$E_{D0}/N_t = \varepsilon_{D0} + J\varepsilon_{\text{AFM}}^{S=1/2}, \tag{26}$$

where $\varepsilon_{\text{AFM}}^{S=1/2}$ ($\varepsilon_{\text{AFM}}^{S=3/2}$) is the ground-state energy per site of the spin-1/2 (spin-3/2) square-lattice Heisenberg AFM with unit couplings. We denote by D0 the doublet with lower energy, $\varepsilon_{D0} \leq \varepsilon_{D1}$, to obtain for the ground state

$$n = \begin{cases} \text{D0} & \text{if } \varepsilon_{D0} + J\varepsilon_{\text{AFM}}^{S=1/2} \leq \varepsilon_Q + J\varepsilon_{\text{AFM}}^{S=3/2}, \\ \text{Q} & \text{otherwise.} \end{cases} \tag{27}$$

The critical inter-trimer coupling where the two levels cross is

$$J_c = \frac{\varepsilon_Q - \varepsilon_{D0}}{\varepsilon_{\text{AFM}}^{S=1/2} - \varepsilon_{\text{AFM}}^{S=3/2}}, \tag{28}$$

and because $J_c > 0$ a first-order quantum phase transition takes place between a spin-1/2 and a spin-3/2 AFM phase. The value of $J_c$ does not depend on $J_2$ because $l_{\Delta,13} = 1$ in both the Q and D0 levels, and the contribution from the $J_2$ dimer cancels in the numerator of Eq. (28).

The reasoning to this point is analogous to the FFB case, with the exception of the additional quantum number, $l_{\Delta,13}$, that labels the doublets D0 and D1. Although the D1 level has no direct influence at $T = 0$, this changes at finite temperatures. When $T > 0$, the first-order transition is expected to continue along a line, $J_c(T)$, up to the critical point at $(T_c, J_c(T_c))$. For $T \ll T_c$, we extend the approximation to finite temperatures by replacing energies with free energies. For non-degenerate D0 and D1, the system will adopt the state with the lower free energy of

$$F_Q/N_t = \varepsilon_Q + J\varepsilon_{\text{AFM}}^{S=3/2}(T/J) - Ts_{\text{AFM}}^{S=3/2}(T/J), \tag{29}$$

$$F_{D0}/N_t = \varepsilon_{D0} + J\varepsilon_{\text{AFM}}^{S=1/2}(T/J) - Ts_{\text{AFM}}^{S=1/2}(T/J). \tag{30}$$

However, if $\epsilon_{D1} - \epsilon_{D0} \lesssim T$, it is not meaningful to consider the configuration $n_\Delta = \text{D0}$ as a macroscopic state, because thermal fluctuations allow admixtures of D0 and D1 to acquire

a finite weight. Instead we subsume all configurations with $n_\Delta \in \{D0, D1\}$ into a single macrostate,

$$e^{-\beta F_D} = \sum_{\substack{\{n_\Delta\} \\ n_\Delta = D0,D1}} e^{-\beta[\varepsilon_n + F(\{n_\Delta\})]} = \left(1 + e^{-\beta(\varepsilon_{D1} - \varepsilon_{D0})}\right)^{N_t} e^{-\beta F_{D0}}, \tag{31}$$

with free energy

$$F_D = F_{D0} - TS_D, \tag{32}$$

$$S_D = N_t \log\left(1 + e^{-\beta(\varepsilon_{D1} - \varepsilon_{D0})}\right). \tag{33}$$

As expected, the additional entropy contribution due to the doublets vanishes when $\varepsilon_{D1} - \varepsilon_{D0} \gg T$ and assumes the maximal value of $N_t \log 2$ in the degenerate case $\varepsilon_{D1} = \varepsilon_{D0}$ ($J_1 = J_2 = J_3$).

These expressions allow us to estimate the first-order line, $J_c(T)$, for $T \ll T_c$ in a manner similar to Ref. [28]. Because $F_Q$ and $F_{D0}$ describe square-lattice Heisenberg AFMs, we employ the standard low-temperature approximation for the specific heat, $C(T,J) = \partial E(T,J)/\partial T/N_t$, to capture the behavior of the free energy at low temperatures,

$$F(T,J) = E(0,J) - TS(0,J) + \int_0^T dT' \left(1 - \frac{T}{T'}\right) N_t C(T',J). \tag{34}$$

For the spin-$S$ AFM, the spin-wave excitations (magnons) give a power-law contribution to the specific heat [57–60],

$$C(T,J) = \frac{\partial J \varepsilon_{AFM}(T/J)}{\partial T} = a(S)\left(\frac{T}{J}\right)^2 + b(S)\left(\frac{T}{J}\right)^4 + \dots, \tag{35}$$

where we have also included the next-to-leading-order correction. The free energy then becomes

$$F(T) = E(0) - TS(0) - \frac{a(S)}{6J^2}T^3 - \frac{b(S)}{20J^4}T^5 + \dots, \tag{36}$$

or for the FFTL model

$$F_Q(T,J)/N_t = \epsilon_Q + J\varepsilon_{AFM}^{S=3/2} - \frac{a(3/2)}{6J^2}T^3 - \frac{b(3/2)}{20J^4}T^5, \tag{37}$$

$$F_{D0}(T,J)/N_t = \epsilon_{D0} + J\varepsilon_{AFM}^{S=1/2} - \frac{a(1/2)}{6J^2}T^3 - \frac{b(1/2)}{20J^4}T^5, \tag{38}$$

$$F_D(T,J)/N_t = F_{D0}(T)/N_t - TS_D/N_t. \tag{39}$$

Values for $a(1/2) = 0.8252$, $a(3/2) = 0.1146$, $b(1/2) = 7.2$, and $b(3/2) = 0.067$ are provided by Ref. [60] and the value $\varepsilon_{AFM}^{S=1/2} = -0.669437(5)$ by Ref. [61]. We did not find a value for $\varepsilon_{AFM}^{S=3/2}$ in the literature, and so we extracted the result $\varepsilon_{AFM}^{S=3/2} = -4.98603(3)$ from our own SSE QMC simulations, as detailed in App. A.

Solving the equation

$$F_Q(T,J_c) = F_D(T,J_c) \tag{40}$$

for different, fixed values of $J_2/J_1$ then yields the transition lines, $J_c(T)$, shown in Fig. 4(e). Direct comparison with the data shown for a finite-sized system in Figs. 4(a-d) confirms that the approximate treatment holds remarkably well in describing the first-order transition line. In fact, within the finite grid of our $(J, T)$ coverage, we do not detect any deviations from it,

even up to $T_c$. However, as discussed earlier, the existence of the critical point is not visible by this approach.

The approximate analysis provides valuable insight into the shapes of the transition lines in Fig. 4(e). In the limit of low temperatures and when $J_2 \neq J_1$, $S_D$ is suppressed exponentially and $J_c(T)$ is dominated by the $T^3$ magnon contribution, leading to the asymptotic behavior

$$J_c(T) \sim J_c + \frac{\varepsilon_{\text{AFM}}^{S=3/2} - \varepsilon_{\text{AFM}}^{S=1/2}}{6(\epsilon_Q - \epsilon_{D0})^2} \left(a(3/2) - a(1/2)\right) T^3, \tag{41}$$

which describes an asymptotically vertical transition line. However, in the special case $J_1 = J_2 = J_3$, $S_D = N_t \log 2$ never becomes irrelevant due to the extensive ground-state degeneracy in the doublet phase. Because $-TS_D$ enters linearly, the result is an asymptotically sloped first-order line,

$$J_c(T) \sim J_c + \frac{\log 2}{\varepsilon_{\text{AFM}}^{S=1/2} - \varepsilon_{\text{AFM}}^{S=3/2}} T, \tag{42}$$

accounting for the clear qualitative difference in lineshapes in Fig. 4(e). A closer inspection of Eq. (36) reveals that obtaining a $T$-linear free-energy contribution, and thus a first-order line with finite slope at $T = 0$, is possible only in the presence such a macroscopic jump in the ground-state entropy.

For intermediate $J_2/J_1$ and at finite temperatures, the doublet entropy term, $-TS_D$, competes with the magnon terms, leading to a smooth crossover between the regime of well separated doublets and the case where they are degenerate (Fig. 4(e)). Even close to $T_c$, the slope of the first-order line is influenced by this competition, allowing us to alter the slope at $T_c$ to a certain extent by changing $J_2/J_1$. We will explore the effects of this parameter in Subsecs. 4.3 and 4.4. However, first we focus on locating the critical point, which is not possible within the approximate analytical approach, and for this we return to QMC simulations and systematic finite-size scaling.

## 4.2 Thermal critical point

As we saw in Fig. 4, the first-order transition line, $J_c(T)$, continues only up to a critical temperature, $T_c$. Because the fluctuations that proliferate at this critical point are those of the scalar $l_\Delta$ degree of freedom, one expects by analogy with the phase diagram of water, and of the FFB [28] and SSL [34], a 2D Ising-type criticality in the FFTL. We now extract $(T_c, J_c(T_c))$ for the different values of $J_2/J_1$ shown in Fig. 4.

Because both the critical temperature, $T_c$, and the critical coupling, $J_c(T_c)$, are unknown, locating the critical point numerically requires the adjustment of two parameters. To accomplish this, we calculated the "quartet susceptibility,"

$$\chi_Q = \frac{\partial^2 f}{\partial \varepsilon_Q^2} = \frac{\beta}{N_t} \left( \langle N_Q^2 \rangle - \langle N_Q \rangle^2 \right), \tag{43}$$

$$= \frac{\beta}{N_t} \left( \left\langle \left( \sum_\Delta l_\Delta \right)^2 \right\rangle - \left\langle \sum_\Delta l_\Delta \right\rangle^2 \right), \tag{44}$$

along lines of fixed $T$. $\chi_Q$ reflects the fluctuations in the number of $l_\Delta = 3/2$ sites, $N_Q$, or equivalently fluctuations in the sum over all $l_\Delta$, and diverges along the first-order line, $J_c(T)$ with $T \leq T_c$, whereas for $T > T_c$ the divergence turns into a finite peak. In a finite system, any divergence at a phase transition is regularized by the system size and also appears as a finite peak. We therefore search for the critical point along the line of susceptibility maxima,

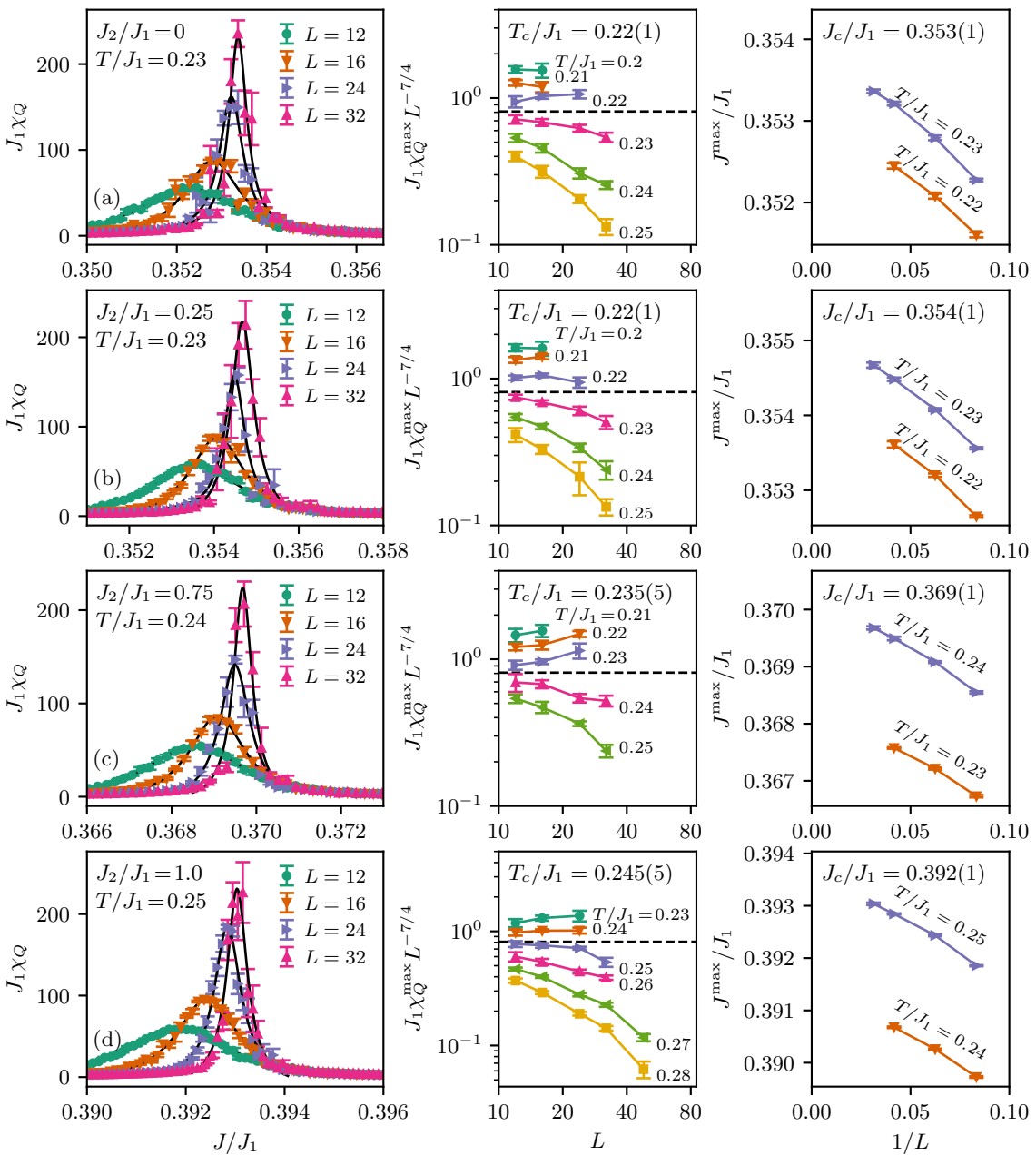

Figure 5: Determination of the critical point of the FFTL model for different intra-trimer couplings, $J_2/J_1$, by finite-size scaling of the maximum in the quartet susceptibility, $\chi_Q(T, J)$, computed along several fixed-$T$ cuts. (left column) $\chi_Q(T \approx T_c, J)$ for different $L$. The maxima are fitted using a Lorentzian (black). (center column) Scaling of the maximum value, $\chi_Q^{\max}$, normalized by the power-law form, $L^{\gamma/\nu} = L^{7/4}$, expected for the 2D Ising universality class. (right column) Scaling of the position, $J^{\max}$, of this maximum.

$(T, J^{\max}(T))$, where the peak height of the susceptibility will follow a (divergent) power-law form as a function of the system size,

$$\chi_Q^{\max}(T_c) \sim L^{\gamma/\nu}, \tag{45}$$

with the critical exponents $\gamma$ and $\nu$. If we assume a 2D Ising critical point, we can find its position by ensuring the best match between the finite-size scaling of the maximum and the exactly known exponents $\gamma = 7/4$ and $\nu = 1$ [62].

In Fig. 5, this program is carried out for different values of $J_2/J_1$. In practice, the simulations become less efficient close to the first-order transition, making the extraction of the maximum, $\chi_Q^{\max}$, intractable for larger system sizes and $T \lesssim T_c$. Nevertheless, by using the known 2D Ising exponents, we are able to extract accurate estimates. Their error is dominated by the uncertainty in $T_c$, which is mostly a consequence of the maximum system sizes we can simulate. We find that $J_c(T_c)$ rises significantly, whereas $T_c$ itself is enhanced only weakly, as $J_2/J_1$ approaches 1. Marking the positions of these critical points in the phase diagrams of Figs. 4(a-d), we find again remarkable agreement between the QMC data and the analytical estimate for the first-order line: the fact that the critical points fall on top of the estimated lines shows that the approximations of Subsec. 4.1 perform well even up to criticality. The rise of $J_c(T_c)$ with $J_2/J_1$ reflects the increasing slope of the first-order lines with $J_2/J_1$, whose thermodynamic consequences we consider next.

## 4.3 Specific heat

The critical point of the FFTL, and of similar models such as the FFB and SSL [28, 34], can be viewed up to the differing dimensionality ($d = 2$ vs. $d = 3$) as a relative to the critical point that terminates the liquid-gas transition line of water. Indeed, these spin models share the critical exponents of the $d = 2$ Ising universality class, in the same way that the critical point of water shows $d = 3$ Ising universality. Further, the critical point in these quantum spin models comes about without the presence of an explicit $Z_2$ symmetry, making them even closer relatives of water than is the $d = 3$ Ising model itself.

At a second glance, however, this correspondence is not trivial. An immediate example arises in a comparison of the specific heat that was performed recently [34]. In the quantum spin systems, the specific-heat data contain two lines of maxima that both approach the critical point sideways relative to the first-order line. While one line of maxima is well known in the specific heat of water, appearing as an extension of the first-order line beyond the critical point and into the supercritical regime, there is no equivalent to the second line of maxima seen in the spin systems. Another distinction is the slope of the first-order line at the critical point with respect to the temperature axis, the FFB and SSL models featuring rather vertical lines quite opposed to the strongly slanted one in water (explicit visualizations may be found in Ref. [34]). These differences raise the general question of the extent to which the two lines of maxima can be a universal feature of the critical point, rather than a model-specific property, and we would like in particular to understand their connection with the slope of the first-order line.

The FFTL, in which $J_2/J_1$ produces quite explicitly the differently slanted first-order transition lines shown in Fig. 4(e), thus provides a valuable model for elucidating this connection. In Fig. 6 we show the specific heat, $C = \partial \langle H \rangle / \partial T$, in the $T$-$J$ plane for 4 different values of $J_2$. In the vicinity of the critical point, we indeed identify two lines of maxima, whose shapes depend on the strength of $J_2$. Although neither the lines nor the actual $C(T)$ curves are ever symmetrical, for $J_2 = 0$ (Figs. 6(a,e)) the lines of maxima do appear at comparable temperatures. This is similar to the specific-heat results for the FFB and SSL models [34], in which we stress that there is no exact symmetry, whereas this symmetry is an exact property of the Ising model (enforced by the $Z_2$ symmetry of the Hamiltonian and displayed in Fig. 7(a)).

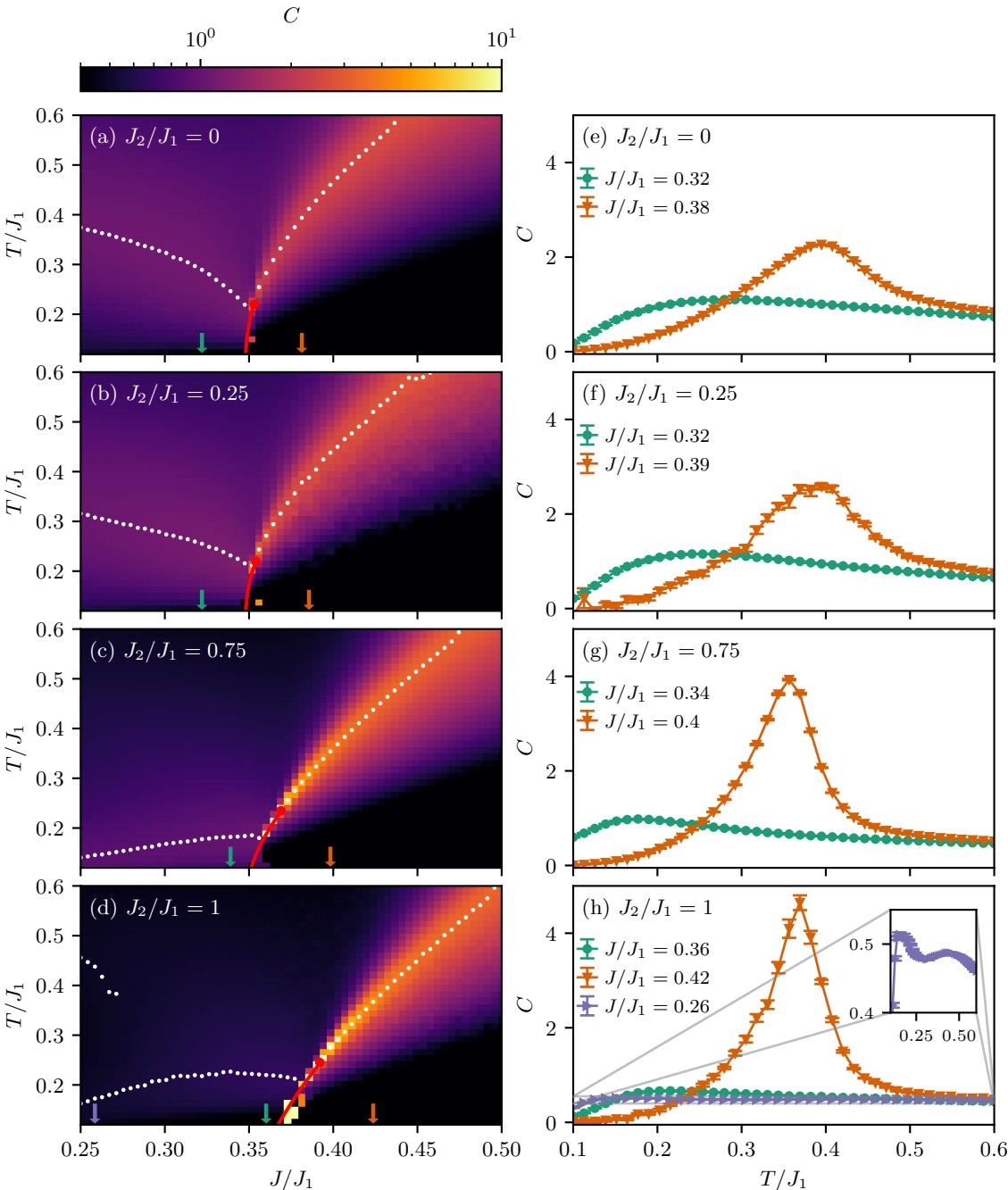

Figure 6: Specific heat, $C(J, T)$, of the $L = 12$ FFTL model for different values of $J_2/J_1$. (a)–(d) $C$ as a function of $J$ and $T$. For each fixed $J$, the temperatures, $T^{\mathrm{max}}$, of the maxima in $C(T)$ are shown as white dots. Red lines and points mark again the approximate first-order line and the numerically determined critical point. Arrows indicate the positions of the fixed-$J$ cuts shown in panels (e)–(h). In panel (h), the inset shows a magnification of the cut at $J/J_1 = 0.26$, highlighting the two-maxima structure.

As $J_2$ is increased and the first-order line slants further to the right, both the temperatures (Figs. 6(b-d)) and peak heights (Figs. 6(f-h)) of the specific heat are suppressed on the left side of the phase diagram ($S = 1/2$ AFM). At $J_2/J_1 = 1$ (Fig. 6(d,h)), the maxima to the left of $J_c$ are barely discernible, least of all when compared to the robust peak in their right-side counterparts, and the situation resembles the specific heat of water, where only one line of maxima is visible. In this way, the FFTL provides a continuous progression from the "Ising-like" to the "water-like" form of the specific heat within a single model, making it a useful example to understand their differences. At the microscopic level, it is the doublet-degeneracy of the FFTL model that allows us to control this progression, and to proceed all the way to the asymmetric, "water-like" form in a manner that was not possible in the FFB and SSL models (which are not far from "Ising-like" in this respect).

Focusing in detail on the positions of the maxima, we observe that only the right-hand line of maxima approaches the critical point directly. The left-hand line of maxima instead meets the first-order line below the critical temperature, $T_c$. This behavior is most obvious around $J_2/J_1 = 1$ and becomes harder to distinguish for smaller $J_2$. However, its presence does point to the fact that the left-hand maxima are not a universal feature of the critical point, and thus it is not surprising that there are models where only one line of maxima appears.

Returning to the analysis of Subsec. 4.1, the specific heat of the FFTL for $J < J_c$ can be understood as the sum of an $S = 1/2$ AFM magnon contribution and the specific heat of decoupled trimers ($J = 0$). The magnon contribution consists of a peak at the temperature scale of $J$, which is independent of the value of $J_2$. Conversely, the trimer contribution is independent of $J$ but depends strongly on $J_2$. When the degeneracy between $\epsilon_{D0}$ and $\epsilon_{D1}$ is broken, the D1 level moves upwards into the gap between $\epsilon_{D0}$ and $\epsilon_Q$, leading for small $J_2/J_1$ to a "doublet peak" that obscures the magnon peak. As $J_2/J_1 \rightarrow 1$, the gap between the doublet levels closes and the associated specific-heat peak moves to lower temperatures, revealing more of the magnon peak. In the degenerate case ($J_2/J_1 = 1$), the doublet entropy is not released at all, leading to a very weak specific-heat signal. In this situation, the magnon and trimer contributions are sufficiently well separated at weak inter-trimer couplings ($J/J_1 \lesssim 0.27$) that they produce two clearly distinguishable, if very weak, local maxima (inset, Fig. 6(h)).

From another perspective, one may ask how much of the specific heat can be understood by considering the free energy close to the critical point, where its singular part will become similar to the part associated with the 2D Ising universality class. This similarity, however, holds only up to a coordinate transformation in the scaling fields, i.e. the quantities known as the temperature and magnetic field in the bare Ising model need not correspond to the physical temperature and couplings of the FFTL.

This type of coordinate transformation not only controls the angle and location of the first-order line in the $(J, T)$ phase diagram, but also has a direct effect on the thermodynamic quantities by transforming and mixing their free-energy derivatives. To elucidate this point, we performed such a transformation explicitly for the specific heat of the bare Ising model on the square lattice,

$$H = -\sum_{\langle i,j \rangle} \sigma_i^z \sigma_j^z - \tilde{h} \sum_i \sigma_i^z, \tag{46}$$

at "bare" temperature $\tilde{T}$ and magnetic field $\tilde{h}$. To compare with the FFTL, which has a variable slanting of the first-order line, we introduce a coordinate transformation that is essentially a rotation around the critical point,

$$\begin{pmatrix} T - T_c \\ h \end{pmatrix} = \begin{pmatrix} \cos(\phi) & -\sin(\phi) \\ \sin(\phi) & \cos(\phi) \end{pmatrix} \begin{pmatrix} \tilde{T} - T_c \\ \tilde{h} \end{pmatrix}, \tag{47}$$

(a related transformation was considered in the context of liquid-liquid transitions in Ref. [63]). Under this transformation, the specific heat mixes with the magnetocaloric correlations and

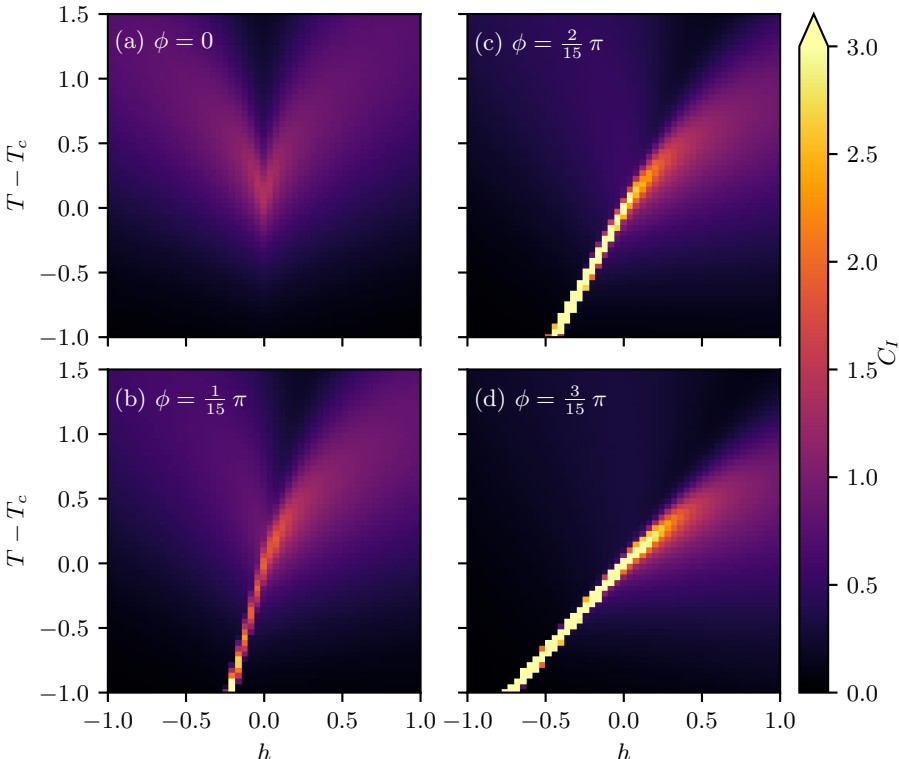

Figure 7: Specific heat, $C_I(T, h)$, of the bare $L = 12$ Ising model in rotated coordinates, shown for different values of the mixing angle, $\phi$, between the bare coordinates $\tilde{T}$ and $\tilde{h}$.

the magnetic susceptibility to give

$$-C_I/T = \frac{\partial^2 f}{\partial T^2} = \cos^2(\phi)\frac{\partial^2 f}{\partial \tilde{T}^2} + \cos(\phi)\sin(\phi)\frac{\partial^2 f}{\partial \tilde{T}\partial\tilde{h}} + \sin^2(\phi)\frac{\partial^2 f}{\partial \tilde{h}^2}. \qquad (48)$$

We evaluate these three free-energy derivatives for the Ising model by computing the correlation functions $\langle H^2 \rangle - \langle H \rangle^2$, $\langle HM \rangle - \langle H \rangle\langle M \rangle$, and $\langle M^2 \rangle - \langle M \rangle^2$, where $M = \sum_i \sigma_i^z$ is the Ising magnetization. For this purpose we employed classical Monte Carlo simulations with Metropolis [64] and Wolff [65] updates using the "ghost spin" method [66] to simulate efficiently in a magnetic field. Figure 7 shows the "rotated" specific-heat results we obtained for different values of the coordinate mixing angle, $\phi$. Looking again at the maxima, we observe that singular contributions appear along the first-order line as soon as $\phi > 0$, and that the right-hand maxima above it are enhanced. The more slanted the first-order line becomes, the more strongly is the right branch enhanced, until even at comparatively small $\phi$ values it dominates the signal completely.

This picture is in good agreement with the specific heat of the FFTL close to the critical point in Fig. 6, where all cases feature slanted first-order lines whose angle increases with their proximity to the degenerate case. Similar behavior is also present in the specific-heat data for the FFB and SSL models [34]. At this point, we stress that the construction of the "rotated" free energy has a meaning only in the vicinity of the critical point: it cannot be written in terms of a partition function of a microscopic model and away from criticality it is the microscopic degrees of freedom that are the most important. Specifically for our discussion of the full first-order lines, their low-temperature shapes are those given by the free-energy arguments of Subsec. 4.1 and shown in Fig. 4(e).

The physics underlying these results is the fact that the most divergent quantity in the

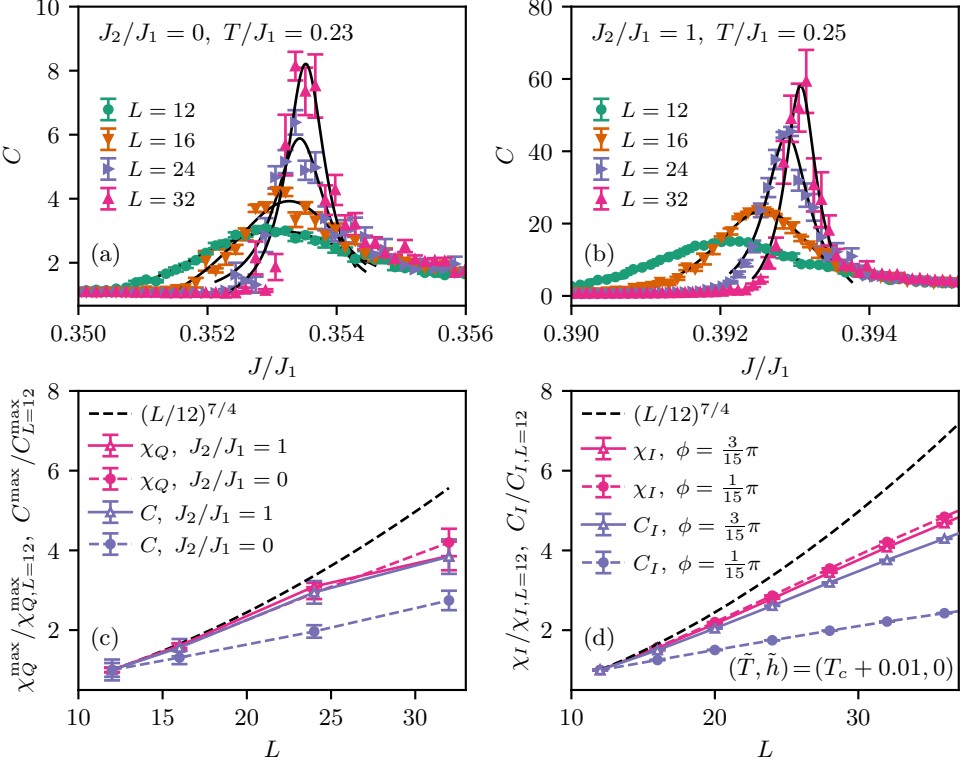

Figure 8: Finite-size scaling of the specific heat, $C(J, T)$, close to the critical point for (a) a non-degenerate FFTL and (b) the degenerate FFTL. (c) Scaling of the specific-heat maxima, $C^{\max}$, shown together with the quartet susceptibility maxima, $\chi_Q^{\max}$, normalized for comparison to their $L = 12$ values. The dashed black line shows a power law with the critical exponent $\gamma/\nu = 7/4$. The temperatures are those of panels (a) and (b). (d) Scaling of the specific heat, $C_I$, and the susceptibility, $\chi_I$, in the rotated Ising model at different mixing angles, $\phi$, normalized to their values with $L = 12$. The data are taken with a small offset from the critical point to mimic the situation in panel (c).

critical scaling of the Ising model is the susceptibility, given by the third term of Eq. (48), whereas the Ising specific heat (first term) has only a weak critical scaling ($\log L$ in 2D). The critical properties of any system described by a rotated Ising model are a mixture of the pure Ising terms, and can therefore be expected to change their form. This behavior is seen most clearly in the critical specific heat of the FFTL, which in contrast to the 2D Ising form grows strongly with system size, as we show in Figs. 8(a-b). While it is difficult to confirm the true asymptotic critical scaling in our FFTL-model data due to the limited accuracy with which we can identify the exact critical point, some useful observations are nevertheless possible. In the degenerate case ($J_2/J_1 = 1$), the scaling of the specific heat coincides with that of the quartet susceptibility up to a normalization factor (Fig. 8(c)). For $J_2/J_1 = 0$, the specific heat does grow more slowly than the quartet susceptiblity, in agreement with the weaker scaling-field mixing of the non-degenerate case, although the susceptibility contribution is still expected to dominate at large system sizes.

A purely Ising-type $\log L$ scaling of the specific heat has been observed at the thermal critical point in the FFB [28], indicating that the mixing of scaling fields is weaker in that model than in the maximally non-degenerate FFTL. For the degenerate FFTL, the match in the scaling behavior of the specific heat and the quartet susceptibility close to critical point can be interpreted thermodynamically as an effect of scaling-field mixing and microscopically by the

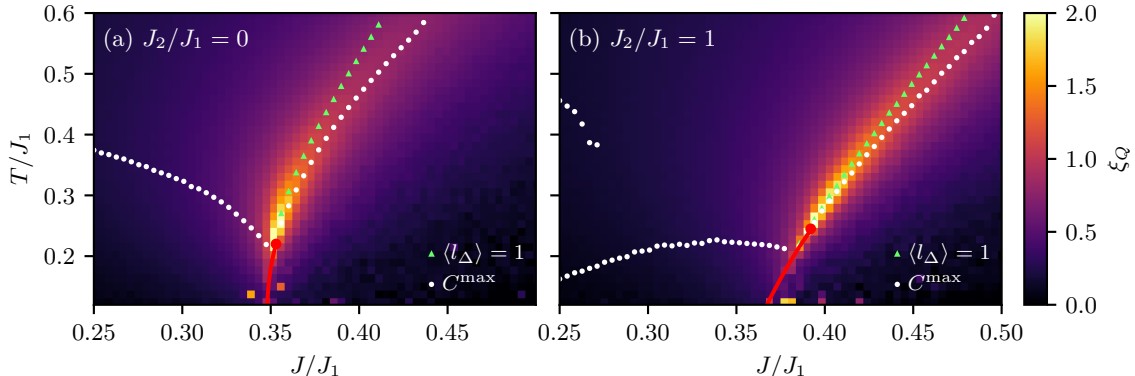

Figure 9: Quartet correlation length, $\xi_Q$, shown as a function of $T$ and $J$ for the $L = 12$ FFTL model at (a) a representative non-degenerate intra-trimer coupling, $J_2/J_1 = 0$, and (b) the degenerate point, $J_2/J_1 = 1$. Superimposed are the lines of constant-$J$ maxima in the specific heat, $T^{\mathrm{max}}$, and the "critical isochore" line where $\langle l_\Delta \rangle = 1$. In both panels, the critical point and the approximate first-order line are shown in red.

fact that the energy fluctuations (reflected in $C$) are dominated by quartet fluctuations ($\chi_Q$) in the absence of a gap between the doublets (Fig. 6(h)). This interpretation is further confirmed by a direct comparison of the model results (Fig. 8(c)) with the scaling of the specific heat in the rotated Ising model (Fig. 8(d)) for different mixing angles. The deviation from the critical $L^{\gamma/\nu}$ scaling in both cases is due to the limited resolution in $(J, T)$ with which we are able to identify the critical point in the FFTL, a situation we introduce in Fig. 8(d) by a deliberate small detuning from the Ising critical temperature, $T_c$.

Finally, in principle one may question the effect of this scaling-field mixing on our estimation of the critical point in Subsec. 4.2, where we compared the growth of the quartet susceptibility directly to the critical scaling of the Ising susceptibility. However, these two quantities no longer have the same direct relationship when the scaling fields become mixed: similar to Eq. (48), the scaling of the singular part of the quartet susceptiblity is given by a combination of Ising observables. However, once again the most divergent contribution is that of the Ising susceptibility, which therefore dominates in the thermodynamic limit, while the other mixed-in contributions behave as subleading corrections to scaling and enter our estimate of the critical point in the same way as additional finite-size effects. The contributions from weaker divergences can be significant in some quantities for the system sizes of our study, most notably for $C(T)$ at smaller values of $J_2/J_1$. However, the results of Fig. 8(c) make clear that the quartet susceptibility, $\chi_Q$, of the FFTL is affected only weakly by the mixing of scaling fields for all values of $J_2/J_1$, making it eminently suitable for a robust extraction of $T_c$ and $J_c(T_c)$. In fact the minor deviation from $L^{\gamma/\nu}$ behavior of $\chi_Q$ is most likely the consequence of a small discrepancy from the exact critical point, which would allow our estimate to be further improved.

## 4.4 Correlation length and characteristic lines

Additional insight into the physics above $T_c$ can be obtained by considering the correlation length associated with the critical point. This we estimate [50] from the quartet structure factor,

$$S_Q(\mathbf{q}) = \frac{1}{N_t} \sum_{\Delta, \Delta'} e^{-i\mathbf{q}\cdot(\mathbf{R}_\Delta - \mathbf{R}_{\Delta'})} (\langle l_\Delta l_{\Delta'} \rangle - \langle l_\Delta \rangle \langle l_{\Delta'} \rangle), \qquad (49)$$

as

$$\xi_Q = \frac{1}{|\delta\mathbf{q}|}\sqrt{\frac{S_Q(\delta\mathbf{q})/S_Q(2\delta\mathbf{q})-1}{4-S_Q(\delta\mathbf{q})/S_Q(2\delta\mathbf{q})}}, \quad \delta\mathbf{q} = \frac{2\pi}{L}\hat{\mathbf{x}}, \tag{50}$$

where $\mathbf{R}_\Delta$ is the position of trimer $\Delta$. Based on this standard Ornstein-Zernike form, we extract the length scale of the critical fluctuations in terms of the corresponding correlation function. We comment here that $S_Q$ encodes the $\langle\mathbf{S}_\Delta^2\mathbf{S}_{\Delta'}^2\rangle$ correlations, which would be an off-diagonal four-spin correlation function in the single-spin $S^z$ basis and thus comparatively difficult to access within SSE QMC. Employing the trimer basis and the methodology of Subsec. 3.2 is therefore beneficial not only in removing the sign problem of the FFTL but also in rendering the correlations crucial to describing its physics diagonal, and as a result easy to compute.

Figure 9 shows scans of $\xi_Q$ across the phase diagram for the two cases $J_2 = 0$ and $J_2/J_1 = 1$. These plots look rather similar to the specific-heat data of Fig. 6, with the notable exception that there is no trace of a second set of maxima to the left of the first-order line. From $\xi_Q$ we therefore identify a single characteristic line within the supercritical regime, which in essence continues the first-order line beyond $T_c$. Along this continuation line, fluctuations in the local trimer states proliferate, anticipating the critical point and then the first-order line that control the low-temperature physics. For comparison, we include in Fig. 9 two other sets of characteristic quantities. One set is the lines of specific-heat maxima from Fig. 6, although for further comparison we neglect the line of weak maxima to the left of $J_c$. The other is the line along which $\langle l_\Delta \rangle = 1$, a value we base on our estimate that $\langle l_\Delta \rangle \approx 1$ at the critical point. By analogy with the density of water at the critical point, we refer to this line as the "critical isochore."

In the supercritical regime, the lines of maxima from these three different thermodynamic quantities converge upon approaching the critical point for any value of $J_2$. Away from criticality, they begin to diverge, each apparently most sensitive to slightly different features of the non-divergent thermal fluctuations. It is notable that both the convergence near the critical point and the proximity of the characteristic lines further from criticality are best in the degenerate limit, $J_2/J_1 = 1$, when the first-order line is most "water-like." Analogous observations regarding the loci of extrema of various response functions were also made in Ref. [63] with a view to determining the Widom line (strictly, the line of zero ordering field) in the critical phenomena of liquid-liquid transitions with differently slanted first-order lines. In this sense, the characteristic lines we have identified may all serve as quantum magnetic analogs of the Widom line, particularly for the degenerate-doublet FFTL.

# 5  Conclusion

We have presented a QMC approach to study highly frustrated quantum spin models in the trimer basis. The abstract directed-loop approach that we introduce requires no system-specific knowledge or bias. It generalizes easily to other computational bases, and thus should be of use for QMC simulations of many other classes of frustrated quantum spin system (for example those composed of four-spin clusters). Here we have applied our algorithm to the $S = 1/2$ Heisenberg antiferromagnet on the FFTL, where it performs efficient and sign-problem-free simulations up to large system sizes, allowing us to investigate the full finite-temperature physics of this model.

A key property of the FFTL model is the appearance of a first-order quantum phase transition between two AFM ground states that break the same symmetry. From the QMC simulations, we found that a line of first-order transitions emerges from this point at finite temperatures, and terminates at a thermal critical point in the 2D Ising universality class. We used a straightforward free-energy argument to understand the shape of the first-order line in terms

of the low-lying excitations and the doublet entropy intrinsic to each trimer. What sets the FFTL apart from the quantum spin models studied to date in this context is the curvature of the first-order line induced by the doublet entropy contribution, which develops in the limit of degenerate doublets into a fully slanting line in the plane of temperature and coupling ratio.

We demonstrated further that this increased slanting of the first-order line is associated with an evolution of the specific heat from the Ising limit, which features two symmetrical lines of maxima, to a single prominent line of maxima that appears as an extension of the first-order line into the supercritical regime. This effect can be explained within the Ising model by a coordinate transformation in the vicinity of the critical point. We computed different characteristic thermodynamic quantities for the FFTL, notably the line of maxima in the correlation length and the critical isochore, to show that they align with the line of primary specific-heat maxima upon approaching the critical point from the supercritical regime, whereas the line of secondary maxima is not related to the critical-point physics. In this way we identified the quantum magnetic analog of the Widom line, serving as a continuation of the first-order line in the supercritical regime.

Our results show that the FFTL model is very valuable for exploring the physics of first-order (quantum) phase transitions and thermal criticality in the absence of symmetry-breaking in frustrated quantum magnets. The model possesses a number of attributes that warrant further investigation, in particular the fact that the first-order quantum transition separates two AFM ground states that break the same symmetry. In principle, this allows quantum fluctuations in an extended phase diagram to terminate the first-order transition at $T = 0$, leading to a quantum critical point. Beyond such a point, one may anticipate a smooth crossover between the $S = 1/2$ and $S = 3/2$ representations or an exotic intermediate phase that mixes the two. It is conceivable that this regime could be reached within the sign-free parameter subspace, for example by introducing a finite magnetic field or an anisotropy in the fully frustrated interactions, both of which would be accessible by the same abstract directed-loop QMC approach that we have developed here.

# Acknowledgements

We thank Fabien Alet for helpful discussions. We acknowledge the support of the Deutsche Forschungsgemeinschaft (DFG, German Research Foundation) through Grant No. WE/3649/4-2 of the program FOR 1807 and through project RTG 1995. We are grateful to the Swiss National Science Foundation, and to the European Research Council (ERC) for funding under the European Union Horizon 2020 research and innovation program (Grant No. 677061). We thank the IT Center at RWTH Aachen University and the JSC Jülich for access to computing time through JARA-HPC.

# A  Ground-state energy of the spin-3/2 Heisenberg model

The free-energy analysis of Subsec. 4.1 requires the ground-state energy per site of the spin-3/2 Heisenberg model on the square lattice. To compute this energy, we performed SSE QMC simulations of the model for several linear system sizes up to $L = 32$ and for inverse temperatures $\beta = 2L$. The AFM ground-state energy at unit coupling in the finite system (following the notation of Sec. 4.1) converges to the thermodynamic limit with the form [57, 61]

$$\varepsilon_{\text{AFM}}^{S=3/2}(L) = \varepsilon_{\text{AFM}}^{S=3/2} + \frac{c}{\beta L^2} + \dots, \tag{51}$$

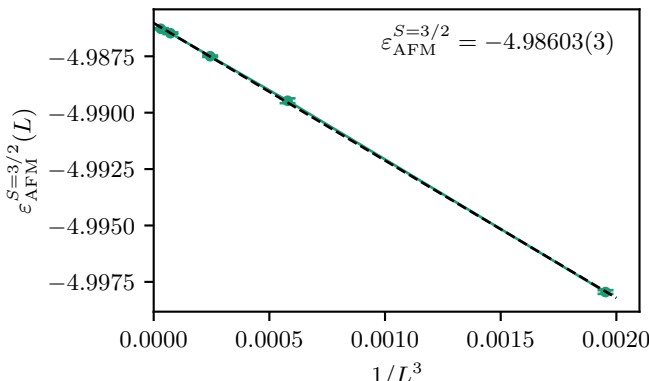

Figure 10: Ground-state energy per site, $\varepsilon_{\text{AFM}}^{S=3/2}(L)$ of the spin-3/2 Heisenberg model. The dashed line shows a fit to Eq. (51), which we use to extract the value in the thermodynamic limit.

where $c$ is a non-universal constant. We use this relation to extract the values $\varepsilon_{\text{AFM}}^{S=3/2} = -4.98603(3)$ and $c = -12.1(1)$ (Fig. 10).

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
