# Peer review of "Quantum Monte Carlo simulations in the trimer basis: first-order transitions and thermal critical points in frustrated trilayer magnets"

_SciPost Physics, doi:SciPost Phys. 12, 054 (2022)_

## Round 2 · Referee Report · Anonymous · 2021-7-20

Strengths
1. Numerical simulations are very clean and well-made
2. Useful results to illustrate why some compounds/models could show two lines of maxima of specific heat, while some others (e.g. water) have only. Connection to recent experiments on frustrated magnets (Ref. 32).
Weaknesses
1. Unclear if the rotated Ising model discussion actually explains the scaling of the specific heat. This should be clarified.
Report
This paper describes quantum Monte Carlo simulations of a frustrated spin system, a trilateral magnet, which have for specificities to have no sign problem in a 3-site (trimer) basis.
At the technical level, this is an extension of previous work (including by the authors) in Refs 23-27 which were mostly concerned with bilayer systems, using a 2-sites basis. There is no major breakthrough here.
Concerning physics, there is a first-order quantum phase transition (driven by the competition between the couplings within or between trimers) similar to the one in the bilayer system, further continued at finite temperature by a line of first-order transitions ending a critical point (of Ising type), similar to the water phase diagram. The main point of the paper is that this frustrated model has an extra tuning parameter (here one of the three couplings inside the trimer) which allows to change the slope of the first-order line, as well as to create/enhance a second maximum in the specific heat (beyond the one existing that continues the first-order line). This second line of maxima is not present in the water phase diagram. This is perhaps not a major physical result, but it is an interesting point that is made, specially in light of recent experiments in SrCu2(BO3)2 (Ref. 32) where the similarity between specific heat data in this frustrated magnet and water was made. This manuscript allows to rationalise (and tune) the different behaviours between models which admit one line of maxima of specific heat, and those which admit two.
The paper is very clear and otherwise well written. I think it satisfies all the criteria for publication in SciPost Physics. Below, I have only one main concern and the rest is mostly simple comments / suggestions.
Requested changes
- My main concern is the discussion on 4.3 using a “rotation” of the critical Ising model to explain the scaling behaviours of specific heat. From my understanding of this discussion, it seems that the specific heat should *always* diverge as L^{gamma / nu} for all models which admit a 2d Ising critical point in an extended phase diagram, except when the Z_2 symmetry is not explicitly broken (phi=0). This sounds a bit surprising, and I wonder whether finite-size scaling of the rotated Ising model (of the kind shown in Figure 7) actually confirm this. My concern is mostly that this perhaps true, but only very close to the critical point. I am not sure this actually explains the scaling reported in Fig 8c, where none of the data scale as expected L^\gamma/nu (neither the quartet susceptibility nor the specific heat). Can the authors report the values of the (quartet) correlation length corresponding to the data in Fig. 8 ? (I assume these must have been obtained). This would allow to see if the data presented there are actually inside the critical region. If this is not the case, I am not sure the full rotated Ising model argument is worth adding to the paper then.
- I would suggest to move the full section 3 into an Appendix, as this is mostly a technical/formal discussion. I don’t think this is a major point of the paper and I also think that quantum Monte Carlo experts (to which perhaps this discussion is aimed at) would not be too much surprised by the abstract directed loop.
- As a minor point, the value of c in Appendix A should be quoted (even if not universal).
Author: Lukas Weber on 2021-08-17 [id 1678]
(in reply to Report 1 on 2021-07-20)We thank the referee very much for their insightful comments. We have attached our detailed response as a pdf file.
Attachment:
response.pdf

---

## Round 3 · Referee Report · Anonymous (Referee 1) · 2021-10-13

Report

The authors have satisfyingly answered the minor points I raised in the first report. This paper is ready for publication.

---

## Round 3 · Referee Report · Anonymous (Referee 2) · 2021-10-14

Strengths

  1. A new QMC algorithm that overcomes the sign problem is certain subclass of frustrated magnets.
  2. Explain the behaviour of specific heat in some frustrated spin systems.

Weaknesses

  1. No real quantum magnet cited that correspond to the model studied.

Report

The authors have studied thermal phase transition in a model of coupled trimers. By switching to a larger trimer basis with an extended Hilbert space, they are able to overcome the sign problem is a wide range of interactions parameters. This is an extension of their earlier work on frustrated bilayer systems.

The authors have also developed a new approach to constructing directed loop updates within the SSE QMC to work in the new basis.

Using the new algorithm, the authors have studied ground state and thermal phase transitions in a system of coupled trimers. An important finding of the study is that by tuning the intra-trimer degree of frustration, the nature of the 1st order thermal transition can be varied over a wide range. This is an interesting result and expands our understanding of the effects of geometric frustration of physical properties of quantum magnets.

The use of expanded local Hilbert space to overcome the sign problem is not new, but any development in this direction is useful as it allows the unbiased simulation of additional class of frustrated systems. The work is valuable for the algorithmic developments as well as the findings related to the specific heat are important to people working in this area. The numerics are sound, the paper is very well written and in the opinion of the present referee, the manuscript deserves to be published. I have only two optional suggestions that the authors might want to address: 1. Can the authors cite some real quantum magnet to which their model is applicable? 2. Can the authors elaborate a little more on how their choice of actions differ from what one would naively expect based on the local Hamiltonian operators by giving specific examples for the case studied?

  • validity: top
  • significance: good
  • originality: ok
  • clarity: top
  • formatting: excellent
  • grammar: perfect

Author:  Lukas Weber  on 2021-10-25  [id 1876]

(in reply to Report 2 on 2021-10-14)

We would like to thank the referee for a careful reading of our manuscript and for the helpful suggestions regarding (1) the connection to real quantum magnets and (2) the direct comparison of the abstract loop actions with local Hamiltonian operators.

(1) The results of our study are expected to serve as a reference for the fate of a first-order transition at finite temperatures when one of the phases has a degenerate or quasi-degenerate ground state, in the same way as the fully frustrated bilayer served as a reference for the non-degenerate case. While a materials realisation of the fully frustrated bilayer is known only for a system with non-Heisenberg spin interactions, it turned out that the same physics is realised almost exactly in the compound SrCu$_2$(BO$_3$)$_2$. We are in the same situation here: although it seems difficult to find a realistic compound with identical inter-trimer bonds in the trilayer structure (or to engineer one using atomically thin magnetic layers), quantum magnets based on triangular motifs with frustrated coupling certainly do exist, and some of these may show part of the same phenomenology covered by varying our $J_2$ parameter when driven to their phase transitions. We have summarised this reasoning and listed some triangle-based materials in Sec. 1 of the revised manuscript. However, we are not currently aware of a triangle-based compound undergoing this type of first-order transition under pressure.

(2) Because the abstract actions act on the local basis states in the same way as simple ket-bra operations, we have added a comparison between a ket-bra operator and the equivalent combination of local spin-1/2 operators. These operators are significantly more complicated than the spin-sum and -difference operators used in the dimer basis and thus they illustrate the increasing complexity of the “physical-operator” picture in higher-dimensional cluster bases. The suggestion of the referee also led us to improve our presentation of the xor actions, clarifying the case where the basis dimension is not a power of two.

---

## Round 3 · Author Response

Dear Editor,

Thank you very much for forwarding us the first referee’s report on our
manuscript “Quantum Monte Carlo simulations in the trimer basis: first-order
transitions and thermal critical points in frustrated trilayer magnets.”
Following your advice, we have improved the manuscript based on the comments
of this referee and we resubmit the revised version for your further action.

We provide a full response to the points raised by the referee and a
summary of changes made in the revision process. This response is accompanied
by two figures.

Best regards,

Lukas Weber, Andreas Honecker, Bruce Normand, Philippe Corboz, Frédéric Mila
and Stefan Wessel

---

## Round 3 · List of Changes

-- three new paragraphs added in the introduction to Sec. 3 to make clear
its embedding as an integral part of the study (critique (2) of the
referee).
-- modified and additional sentences included in Secs. 1, 2, 4, and the abstract to assist
with the embedding of Sec. 3.
-- additional panel in Fig. 8 and additional sentences in the accompanying
text (Sec. 4.3) concerning how criticality is reflected in the data
shown (from critique (1) of the referee).
-- add subscript $I$ to the Ising specific heat, $C_I$ for clarity.
-- subscript “c” corrected in the y-axis labels of Fig. 7.
-- missing square added in the denominator of Eq. (39).
-- missing index $i$ added in the definition of the Ising magnetization.
-- citation added to a recent reference concerning the sign problem.

---

## Round 4 · Author Response

Dear Editor,

Thank you very much for forwarding the second round of referee reports for our manuscript "Quantum Monte Carlo simulations in the trimer basis: first-order transitions and thermal critical points in frustrated trilayer magnets." Thank you in advance for your further consideration of our contribution.

Best regards,

Lukas Weber, Andreas Honecker, Bruce Normand, Philippe Corboz, Frédéric Mila and Stefan Wessel

---

## Round 4 · List of Changes

We have addressed the two suggestions by the new referee (i) in a direct response and (ii) in the revised manuscript. Indeed these points help to improve the quality of the manuscript, which we resubmit here.

---

## Editorial Decision

published